# Collaborative Optimal Formation Control for Heterogeneous Multi-Agent Systems

**DOI:** 10.3390/e24101440

**Published:** 2022-10-10

**Authors:** Yandong Li, Meichen Liu, Jiya Lian, Yuan Guo

**Affiliations:** College of Computer and Control Engineering, Qiqihar University, Qiqihar 161000, China

**Keywords:** heterogeneous multi-agent system, unmanned aerial vehicle, unmanned ground vehicle, optimal control, cooperative formation control

## Abstract

In this paper, the distributed optimal control method is used to study the cooperative formation of heterogeneous multi-agents in the air–ground environment. The considered system consists of an unmanned aerial vehicle (UAV) and an unmanned ground vehicle (UGV). The optimal control theory is introduced into the formation control protocol, the distributed optimal formation control protocol is designed, and the stability is verified by graph theory. Furthermore, the cooperative optimal formation control protocol is designed, and the stability is analyzed using a block Kronecker product and matrix transformation theory. Through the comparison of simulation results, the introduction of optimal control theory shortens the formation time of the system and accelerates the convergence speed of the system.

## 1. Introduction

With the rapid development of science and technology, the world’s military powers attach great importance to the cooperation capability of an unmanned combat system. In recent decades, air–ground heterogeneous unmanned combat systems, which consist of unmanned aerial vehicles (UAVs) and unmanned ground vehicles (UGVs), have been favored by military powers due to their fast response speed, strong communication capability, strong payload capacity, and high target reconnaissance accuracy [1,2,3].

In the field of multi-agent systems, cooperative control has received extensive attention and research. Examples of such applications include the field of robot collaboration, UAV formation, cooperative transport, and combat reconnaissance [4]. Formation control is an application hotspot in the field of distributed cooperation. In general, formation control can be divided into two categories according to the presence or absence of leader agents: leader and leaderless [5,6,7,8]. Reference [9] used the leader–follower method to complete the trajectory tracking task of the UAV–UGV system but did not cooperate in the process of completing the formation. Furthermore, Reference [10] studied the cooperation problem of the UAV–UGV system by improving the artificial physics approach but did not study the formation problem. Reference [11] studied the time-varying formation control of cooperative heterogeneous multi-agent systems and combined formation with cooperation.

In the cooperative reconnaissance and cooperative strike of the unmanned combat system, based on forming an established formation, the unmanned combat system must also identify the complex and volatile battlefield environments and quickly cross barriers, such as fences and fortifications. Therefore, quickly making the whole formation reach the desired state is also an important concern of the formation problem. Reference [12] used a virtual-structure-based approach and multiple-impedance control to achieve the optimal formation of three mobile robots, and the mobile robots carried out the cooperative formation. However, this study was based on the study of homogeneous multi-agents. In Reference [13], the leader–follower strategy and the virtual leader strategy were integrated into an optimal control framework to study the optimal formation of multiple UAVs. However, this study did not investigate the cooperation of multiple UAVs. For the optimal formation of heterogeneous multi-agent, there are few related research results. Reference [14] is based on using reinforcement learning methods to achieve the optimal formation of heterogeneous multi-agent systems but does not study cooperative control. It should be noted that optimal formation control alone can only solve a relatively limited number of problems, and cooperative optimal formation control is still an open problem.

In addition, most of the existing research results are based on the same dynamic model, namely the homogeneous agent model [15,16,17,18]. Compared with homogeneous multi-agent systems, multi-agent systems composed of heterogeneous dynamic models are more flexible in practical applications. Therefore, it is of great significance to study heterogeneous multi-agent systems. A large number of valuable research results have been obtained for the heterogeneous cooperative problem [19,20,21].

In this paper, the cooperative optimal formation problem of heterogeneous multi-agent systems is studied on the unmanned aerial vehicle and unmanned ground vehicle model. There are threefold main innovations:

Firstly, a heterogeneous modeled UAV–UGV system is proposed, a cooperative architecture of heterogeneous multi-agent systems with equal number is designed, and a Laplacian matrix of communication topology is designed. In addition, a novel block Kronecker product is used to describe the UAV–UGV system. Based on this, distributed formation control is proposed.

The second contribution is to introduce the optimal control method into the formation control protocol, design the distributed formation optimal control protocol, and prove the stability using the method of graph theory.

The third contribution is to design a cooperative formation control protocol for the air–ground system based on the heterogeneous system model so that the UAV–UGV system can achieve the cooperative formation effect. Then, the optimal control is introduced into the cooperative formation control protocol, and the cooperative optimal formation control protocol is designed, which enables the UAV–UGV system formation to quickly achieve the expected effect.

## 2. Preliminaries

This section mainly introduces the preliminary knowledge of unmanned aerial vehicles and unmanned ground vehicles, including the use of graph theory to describe the internal relationship of the system and the state-space equation of the UAV system and the UGV system.

### 2.1. Graph Theory

A weighted undirected graph G=(V,E,A) consists of n vertices, where $V=\left(v_{1},v_{2},\cdots v_{n}\right)$ represents the set of all vertices in the undirected graph, and each vertex represents an agent. $E=\left\{e_{ij}=\left(v_{i},v_{j}\right)\right\}\subseteq v\times v$ represents the edge set between vertices, and $e_{ij}=\left(v_{i},v_{j}\right)$ represents the edge from $v_{i}$ vertex to $v_{j}$ vertex; an edge connection between two vertices indicates that there is an information interaction between these two vertices. The graph is undirected if it allows two-way communication; otherwise, it is directed.$\;A={\left[a_{ij}\right]}_{n\times n}$ represents the adjacency matrix indicating the relationship between agents, where $a_{ij}$ is the weight of the side $e_{ij}=\left(v_{i},v_{j}\right)$ and where the diagonal elements of the matrix A are all 0. For i,j=1,2,3,⋯,n(i≠j), if the agents $v_{i}$ and $v_{j}$ can receive information from each other, then the elements in the adjacency matrix are $a_{ij}=a_{ji}>0$; otherwise, the element in the adjacency matrix is 0.

In an undirected graph, the degree represents the number of neighbors of a node, that is, the number of edges per node. $D=dig\left\{d_{1},\ldots ,d_{n}\right\}$ of undirected graph G is a diagonal matrix with $d_{i}=\overset{n}{\underset{j=1}{\sum }}a_{ij}$.Then, the Laplacian matrix of G is defined as *L* = *D*
−
*A*, which has at least one zero eigenvalue with $1={\left[1,1\ldots ,1\right]}^{T}$ as its corresponding right eigenvector. In addition, *L* has exactly one zero eigenvalue if and only if the directed graph G contains a directed spanning tree.

### 2.2. UGV Dynamics Model

Single UGV motion model:(1)$$\{\begin{array}{l} {\dot{p}}_{gi}=v_{gi}\\ {\dot{v}}_{gi}=u_{gi} \end{array}$$
where $\;p_{gi}={\left[p_{gi}^{x},p_{gi}^{y},p_{gi}^{z}\right]}^{T}$ stands for the position in ground space, $v_{gi}={\left[v_{gi}^{x},v_{gi,}^{y},v_{gi}^{z}\right]}^{T}$ is the velocity in the direction $p_{gi}$, and $u_{gi}={\left[u_{gi}^{x},u_{gi}^{y},u_{gi}^{z}\right]}^{T}$ represents the input of agent i. If there are k UGVs, the above formula is converted to the states:(2)$${\dot{X}}_{G}=A_{G}X_{G}+B_{G}U_{G}$$
where $\;X_{G}={\left(P_{G},V_{G}\right)}^{T}$,$P_{G}=\left(p_{1},p_{2},p_{3}\ldots p_{k}\right)$,$p_{i}=\left(x_{i},y_{i},z_{i}\right),i=1,2,\ldots ,k$;
$$V_{G}=\left(v_{1},v_{2},v_{3}\ldots v_{k}\right),v_{i}=\left(v_{i}^{x},v_{i}^{y},v_{i}^{z}\right),i=1,2,\ldots ,k$$
$$U_{G}=\left(u_{1},u_{2},u_{3}\ldots u_{k}\right),u_{i}=\left(u_{i}^{x},u_{i}^{y},u_{i}^{z}\right),i=1,2,\ldots ,k$$
$$A_{G}=\left[\begin{array}{cc} 0 & 1\\ 0 & 0 \end{array}\right]⊗I_{k},B_{G}=\left[\begin{array}{c} 0\\ 1 \end{array}\right]⊗I_{k}.$$Subscript G represents the state variable of the unmanned vehicle.

The expected formation state is $h_{g}={\left(h_{g}^{x},h_{g}^{y},h_{g}^{z}\right)}^{T}$, the formation state is transformed into the position state, and the new error position state naturally appears, namely:$\;\delta_{G}={\left(\delta_{G}^{x},\delta_{G}^{y},\delta_{G}^{Z}\right)}^{T}={\left(P_{gi}^{x}-h_{g}^{x},P_{gi}^{y}-h_{g}^{y},P_{gi}^{z}-h_{g}^{z}\right)}^{T}$.

Therefore, the problem of formation control becomes finding a protocol $U_{G}$ to drive the error vector $\delta_{G}$ to zero, which means that
(3)$${lim}_{t\to \infty }^{\;}∥\delta_{gi}-\delta_{gj}∥=0,{lim}_{t\to \infty }^{\;}=∥v_{gi}∥=0.$$

### 2.3. UAV Dynamics Model

The motion model of a single UAV is:(4)$$\{\begin{array}{l} \ddot{x}=g\theta \\ \ddot{y}=-g\phi \\ \ddot{z}=f_{z}/m-g\\ \ddot{\phi }=M_{\phi }/I_{x}\\ \ddot{\theta }=M_{\theta }/I_{y}\\ \ddot{\varphi }=M_{\varphi }/I_{z} \end{array}$$
where  g is the acceleration of gravity; x,y, and z are the positions of the UAV in three coordinate systems; ϕ,θ, and φ are the roll angle, the pitch angle, and the yaw angle of the UAV, respectively; $f_{z}$ is the lift force in the direction of height; $M_{\phi },M_{\theta },$ and $M_{\varphi }$ are the torques on the three axes of the body coordinate system; $I_{x},I_{y}$, and $I_{Z}$ are the inertial matrices in the body coordinate system. For L UAVs, the above equations are converted to the state-space form as follows:(5)$${\dot{X}}_{A}=A_{A}X_{A}+B_{A}U_{A}$$
where $X_{A}={\left(P_{A},V_{A},\Omega_{A},{\dot{\Omega }}_{A}\right)}^{T}$,$P_{A}=\left(p_{1},p_{2},p_{3}\ldots p_{l}\right)$,$p_{i}=\left(x_{i},y_{i},z_{i}\right),i=1,2,\ldots ,l$;
$$V_{A}=\left(v_{1},v_{2},v_{3}\ldots v_{l}\right),v_{i}=\left(v_{i}^{x},v_{i}^{y},v_{i}^{z}\right),i=1,2,\ldots ,l$$
$$\Omega_{A}=\left(\Omega_{1},\Omega_{2},\Omega_{3},\ldots \Omega_{l}\right),\Omega_{i}=\left(g\theta_{i},-g\phi_{i},0\right),i=1,2,\ldots ,l$$
$${\dot{\Omega }}_{A}=\left({\dot{\Omega }}_{1},{\dot{\Omega }}_{2},{\dot{\Omega }}_{3},\ldots {\dot{\Omega }}_{l}\right),{\dot{\Omega }}_{i}=\left(g{\dot{\theta }}_{i},-g{\dot{\phi }}_{i},0\right),i=1,2,\ldots ,l$$
$$A_{A}=\left(\begin{array}{cccc} 0 & 1 & 0 & 0\\ 0 & 0 & 1 & 0\\ 0 & 0 & 0 & 1\\ 0 & 0 & 0 & 0 \end{array}\right)⊗I_{l},B_{A}=\left(\begin{array}{c} 0\\ 0\\ 0\\ 1 \end{array}\right)⊗I_{l}$$
$$U_{A}={\left(u_{1},u_{2},u_{3}\ldots u_{l}\right)}^{T},u_{i}=\left(u_{i}^{x},u_{i}^{y},u_{i}^{z}\right),i=1,2,\ldots ,l.$$Subscript A represents the state variable of the unmanned vehicle.

The expected formation state is $h_{a}={\left(h_{a}^{x},h_{a}^{y},h_{a}^{z}\right)}^{T}$, the formation state is transformed into the position state, and the new error position state naturally appears, namely: $\delta_{A}={\left(\delta_{A}^{x},\delta_{A}^{y},\delta_{A}^{z}\right)}^{T}={\left(P_{ai}^{x}-h_{a}^{x},P_{ai}^{y}-h_{a}^{y},P_{ai}^{z}-h_{a}^{z}\right)}^{T}$.

Therefore, the problem of formation control becomes finding a protocol $U_{A}$ to drive the error vector $\delta_{A}$ to zero, which means that
(6)$${lim}_{t\to \infty }^{\;}∥\delta_{ai}-\delta_{aj}∥=0,{lim}_{t\to \infty }^{\;}=∥v_{ai}∥=0,{lim}_{t\to \infty }^{\;}∥\Omega_{i}∥=0,{lim}_{t\to \infty }^{\;}∥{\dot{\Omega }}_{i}∥=0.$$

### 2.4. Heterogeneous Multi-Agent System

To analyze heterogeneous multi-agent systems more conveniently, the UAV system and UGV system are written into the same state space and combined with the state-space model of the single agent above; the form of the heterogeneous multi-agent state-space model is defined as:(7)$$\dot{X}=AX+BU$$
where $X={\left(X_{G}^{T},X_{A}^{T}\right)}^{T},A=\left(\begin{array}{cc} A_{G} & 0\\ 0 & A_{A} \end{array}\right),B=\left(\begin{array}{cc} B_{G} & 0\\ 0 & B_{A} \end{array}\right),\;\mathrm{and}\;U={\left(U_{G}^{T},U_{A}^{T}\right)}^{T}$. The Laplace matrix is $L=\left(\begin{array}{cc} L_{AA} & L_{AG}\\ L_{GA} & L_{GG} \end{array}\right)$, where $L_{AG},L_{GA}$ represents information between heterogeneous agent systems. This paper takes the heterogeneous multi-agent system composed of three UGVs and three UAVs as the research object, and its Laplace matrix relationship is as follows:$$L=\left(\begin{array}{cc} \begin{array}{c} \underset{L_{AA}}{\underbrace{\begin{array}{cc} \begin{array}{c} -3\\ \begin{array}{c} 1\\ 1 \end{array} \end{array} & \begin{array}{cc} \begin{array}{c} 1\\ \begin{array}{c} -3\\ 1 \end{array} \end{array} & \begin{array}{c} 1\\ \begin{array}{c} 1\\ -3 \end{array} \end{array} \end{array} \end{array}}}\\ \underset{L_{GA}}{\underbrace{\begin{array}{cc} \begin{array}{c} \begin{array}{c} 0\\ 0 \end{array}\\ 1 \end{array} & \begin{array}{cc} \begin{array}{c} \begin{array}{c} 0\\ 1 \end{array}\\ 0 \end{array} & \begin{array}{c} \begin{array}{c} 1\\ 0 \end{array}\\ 0 \end{array} \end{array} \end{array}}} \end{array} & \begin{array}{c} \underset{L_{AG}}{\underbrace{\begin{array}{cc} \begin{array}{cc} \begin{array}{c} \begin{array}{c} 0\\ 0 \end{array}\\ 1 \end{array} & \begin{array}{c} \begin{array}{c} 0\\ 1 \end{array}\\ 0 \end{array} \end{array} & \begin{array}{c} \begin{array}{c} 1\\ 0 \end{array}\\ 0 \end{array} \end{array}}}\\ \underset{L_{GG}}{\underbrace{\begin{array}{cc} \begin{array}{c} \begin{array}{c} -3\\ 1 \end{array}\\ 1 \end{array} & \begin{array}{cc} \begin{array}{c} \begin{array}{c} 1\\ -3 \end{array}\\ 1 \end{array} & \begin{array}{c} \begin{array}{c} 1\\ 1 \end{array}\\ -3 \end{array} \end{array} \end{array}}} \end{array} \end{array}\right)$$

The expected formation state is $h={\left(h_{a}^{T},h_{g}^{T}\right)}^{T}$, the formation state is transformed into the position state, and the new error position state naturally appears, namely:$$\delta ={\left(\delta_{A}^{x},\delta_{A}^{y},\delta_{A}^{Z},\delta_{G}^{x},\delta_{G}^{y},\delta_{G}^{Z}\right)}^{T}={\left(P_{ai}^{x}-h_{a}^{x},P_{ai}^{y}-h_{a}^{y},P_{ai}^{z}-h_{a}^{z},P_{gi}^{x}-h_{g}^{x},P_{gi}^{y}-h_{g}^{y},P_{gi}^{z}-h_{g}^{z}\right)}^{T}.$$

Therefore, the problem of formation control becomes finding a protocol U to drive the error vector δ to zero, which means that
(8)$$\underset{t\to \infty }{lim}∥\delta_{i}-\delta_{j}∥=0,\underset{t\to \infty }{lim}=∥v_{i}∥=0,\underset{t\to \infty }{lim}∥\Omega_{i}∥=0,\underset{t\to \infty }{lim}∥{\dot{\Omega }}_{i}∥=0$$


## 3. Design of Control Protocol

To realize the formation of heterogeneous multi-agent systems of UAVs and UGVs, this section is based on the formation control protocol. Firstly, the optimal control law is applied to the single agent. Then, according to the combination of the optimal control law and the formation control protocol, a heterogeneous multi-agent system with distributed optimal formation control is realized. Finally, according to the motion equation of the heterogeneous multi-agent system, the cooperative formation control and cooperative optimal formation control of the heterogeneous multi-agent system are realized.


**Lemma** **1.**[22]. *For an*
N*N
*Laplacian matrix*
$L,Ne^{-Lt},t>0$
*is a random matrix with positive diagonal elements. If*
L
*has a unique zero eigenvalue, Rank*
(N)=N−1*, then its left eigenvector has*
$v={\left[v_{1},v_{2},\cdots v_{n}\right]}^{T}\geq 0$
*and*
$1_{N}^{T}v=1,\;L^{T}v=0$*, where,*
t→∞*,*$e^{-Lt}\to 1_{N}v^{T}$.


### 3.1. Formation Control

Formation control protocol for the UAVs:
(9)$$u_{ia}=\alpha \sum_{j\in N_{i}}^{\;}\left(\delta_{aj}-\delta_{ai}\right)-\beta v_{ai}-\gamma_{1}\Omega_{i}-\gamma_{2}{\dot{\Omega }}_{i}.$$

Formation control protocol for the UGVs:
(10)$$u_{ig}=\alpha \sum_{j\in N_{i}}^{\;}\left(\delta_{gj}-\delta_{gi}\right)-\beta v_{gi}$$ where $\alpha ,\beta ,\gamma_{1},\gamma_{2}$ represent the positive gain coefficients,
$$\delta_{ai}={\left(\delta_{ai}^{x},\delta_{ai}^{y},\delta_{ai}^{z}\right)}^{T},\;\delta_{gi}={\left(\delta_{gi}^{x},\delta_{gi}^{y},\delta_{gi}^{z}\right)}^{T},\;v_{gi}={\left({\dot{P}}_{gi}^{x},{\dot{P}}_{gi}^{y},{\dot{P}}_{gi}^{z}\right)}^{T},\;v_{ai}={\left({\dot{P}}_{ai}^{x},{\dot{P}}_{ai}^{y},{\dot{P}}_{ai}^{z}\right)}^{T}$$
$$\Omega_{i}={\left(g\theta ,-g\phi ,0\right)}^{T},{\dot{\Omega }}_{i}={\left(g\dot{\theta },-g\dot{\phi },0\right)}^{T}$$

Protocols (9) and (10) shall be unified into the same type:(11)$$U=-L_{d}\cdot \tilde{X}$$

Define the state-space form of the multi-agent system formation:(12)$$\dot{\tilde{X}}=A\tilde{X}+BU$$
where
$$L_{d}=(\begin{array}{llllll} \alpha L_{s}⊗I_{3} & -\beta I_{l}⊗I_{3} & -\gamma_{1}I_{l}⊗I_{3} & -\gamma_{2}I_{l}⊗I_{3} & 0 & 0\\ 0 & 0 & 0 & 0 & \alpha L_{s}⊗I_{3} & -\beta I_{k}⊗I_{3} \end{array}),$$

$L_{S}=\left(\begin{array}{lll} -1 & 1 & 0\\ 0 & -1 & 0\\ 1 & 0 & -1 \end{array}\right),\;\tilde{X}={\left(\delta_{ai}^{T},v_{ai}^{T},\Omega^{T},{\dot{\Omega }}^{T},\delta_{gi}^{T},v_{gi}^{T}\right)}^{T},A=\left(\begin{array}{cc} A_{A} & 0\\ 0 & A_{G} \end{array}\right),\;B=\left(\begin{array}{cc} B_{A} & 0\\ 0 & B_{G} \end{array}\right),$I is the identity matrix, and ⊗ is the Kronecker product.


**Theorem** **1.***If Protocols (9) and (10) are satisfied*$\alpha >0,\beta >0,\gamma_{1}>0,\gamma_{2}>0$*,*$\beta \gg \alpha ,\gamma_{1}>\beta ,\gamma_{2}>\beta ,\beta \gamma_{1}\gamma_{2}>\beta^{2}\gamma_{2}\alpha$*, Systems (2) and (5) can implement the formation defined in (3) and (6)*.


**Proof of Theorem** **1.**Substitute Formula (11) into Formula (12) to obtain: $\dot{\tilde{X}}=-T_{d}\cdot \tilde{X}$ where $T_{d}=\left[\begin{array}{llllll} 0 & 1 & 0 & 0 & 0 & 0\\ 0 & 0 & 1 & 0 & 0 & 0\\ 0 & 0 & 0 & 1 & 0 & 0\\ \alpha L_{s}⊗I_{3} & -\beta I_{l}⊗I_{3} & -\gamma_{1}I_{l}⊗I_{3} & -\gamma_{2}I_{l}⊗I_{3} & 0 & 0\\ 0 & 0 & 0 & 0 & 0 & 1\\ 0 & 0 & 0 & 0 & \alpha L_{s}⊗I_{3} & -\beta I_{k}⊗I_{3} \end{array}\right]$According to the linear stability theorem, the parameters $\alpha ,\beta ,\gamma_{1}$, and $\gamma_{2}$ need to be selected so that $T_{d}$ has a zero eigenvalue and other eigenvalues have genuine negative parts. The parameters α and β need to meet the stability of UGV consistency, and the parameters $\gamma_{1}$ and $\gamma_{2}$ need to meet the stability of UAV consistency. After selecting parameters, $T_{d}$ can be converted to a Jordan standard type: $T_{d}=PJP^{-1}$. Let $v_{1}^{T}$ be the first row of $P^{-1}$ and the left eigenvector have eigenvalue 0. Let $w_{1}\;\mathrm{be}$ the first column of P and the right eigenvector have eigenvalue 0. Therefore, $v_{1}^{T}w_{1}=1$; as time approaches infinity, the system’s state becomes:. According to Lemma 1, as time approaches infinity, Systems (2) and (5) asymptotically agree and the systems complete formation. □

### 3.2. Optimal Control

The solution of optimal control requires the states of all multi-agents. Before providing performance indicators, $\overset{¯}{X}$ and $\overset{¯}{W}$ are defined:

$\overset{¯}{X}={\left(\delta_{ai}^{x},\delta_{ai}^{y},\delta_{ai}^{z},{\dot{P}}_{ai}^{x},{\dot{P}}_{ai}^{y},{\dot{P}}_{ai}^{z},g\theta ,-g\phi ,0,g\dot{\theta },-g\dot{\phi },0\right)}^{T},\overset{¯}{W}={\left(\delta_{gi}^{x},\delta_{gi}^{y},\delta_{gi}^{z},{\dot{P}}_{gi}^{x},{\dot{P}}_{gi}^{y},{\dot{P}}_{gi}^{z}\right)}^{T}$ Then, define the performance indicator function as: 



(13)
$$J_{i}=\int_{0}^{\infty }\left[{\bar{X}}_{i}^{T}Q{\bar{X}}_{i}+u_{ia}^{T}Ru_{ia}\right]dt$$





(14)
$$\omega_{i}=\int_{0}^{\infty }\left[{\bar{w}}_{i}^{T}Q{\bar{w}}_{i}+u_{ig}^{T}\gamma u_{ig}\right]dt$$



As the UAV and the UGV are independent in different coordinate systems, the UAV weight must be set to $Q=q^{*}I_{12},R=r*I_{3},\;$ where, q>0, r>0. The UGV weight must be set to $T=\lambda *I_{6},\gamma =\mu *I_{3}$, where λ>0,μ>0.

According to the optimal control theory, the optimal control law of a single agent UAV is: $u_{a}^{*}=-R^{-1}B_{A}^{T}P_{A}\overset{¯}{X}$, where $P_{A}$ is the solution of Riccati’s in Equation (15)
(15)$$A_{A}^{T}P_{A}+P_{A}A_{A}-P_{A}B_{A}R^{-1}B_{A}^{T}P_{A}+Q=0$$

The optimal control law for a single agent UGV is: $u_{g}^{*}=-\gamma^{-1}B_{G}^{T}P_{G}\overset{¯}{w}$, where $P_{G}$ is the solution of Riccati’s in Equation (16)
(16)$$A_{G}^{T}P_{G}+P_{G}A_{G}-P_{G}B_{G}\gamma^{-1}B_{G}^{T}P_{G}+T=0$$

Through the above calculation, the optimal control law $u_{a}^{*}$ can be obtained. Let $K=R^{-1}B_{A}^{T}P_{A}$, the dimension of matrix K is 3∗12,K expressed as $K=\left[k_{1},k_{2},k_{3},k_{4}\right]⊗I_{3}.$ Similarly, the optimal control law $u_{g}^{*}$ can also be solved, let $G=\gamma^{-1}B_{G}^{T}P_{G}$ the dimension of matrix G is 3∗6,G expressed as $G=\left[g_{1},g_{2}\right]⊗I_{3}$.

### 3.3. Distributed Optimal Formation Control

All UAVs have the same dynamics model, so all UAVs are homogeneous multi-agents. Similarly, all UGVs are homogeneous multi-agents. Therefore, optimal control laws can be extended to the formation control of UAV and UGV multi-agent systems.



(17)
$$u_{ia}^{*}=k_{1}\sum_{j\in N_{i}}^{\;}\left(\delta_{aj}-\delta_{ai}\right)-k_{2}v_{ai}-k_{3}\Omega_{i}-k_{4}{\dot{\Omega }}_{i}$$





(18)
$$u_{ig}^{*}=g_{1}\sum_{j\in N_{i}}^{\;}\left(\delta_{gj}-\delta_{gi}\right)-g_{2}v_{gi}$$



Define the multi-agent system to be optimized:(19)$$\dot{\tilde{X}}=A\tilde{X}+BU$$
where $\tilde{X}={\left(\delta_{ai}^{T},v_{ai}^{T},\Omega^{T},{\dot{\Omega }}^{T},\delta_{gi}^{T},v_{gi}^{T}\right)}^{T},A=\left(\begin{array}{cc} A_{A} & 0\\ 0 & A_{G} \end{array}\right),\;B=\left(\begin{array}{cc} B_{A} & 0\\ 0 & B_{G} \end{array}\right)$,$k_{1},k_{2},k_{3},$ and $k_{4}$ are derived from the matrix $K.g_{1}$ and $g_{2}$ are derived from the matrix G.


**Theorem** **2.**
*If the unmanned ground vehicle system in (2) and the unmanned aerial vehicle system in (5) use Protocols (17) and (18), respectively, the formation can be completed, and Performance Functions (13) and (14) can be optimized.*



**Proof of Theorem** **2.**Protocols (17) and (18) shall be unified into the same type:(20)$$U=-L_{l}\cdot \tilde{X}$$
where
$$L_{l}=(\begin{array}{llllll} k_{1}L_{s}⊗I_{3} & -k_{2}I_{l}⊗I_{3} & -k_{3}I_{l}⊗I_{3} & -k_{4}I_{l}⊗I_{3} & 0 & 0\\ 0 & 0 & 0 & 0 & g_{1}L_{s}⊗I_{3} & -g_{2}I_{k}⊗I_{3} \end{array}),$$$L_{S}=\left(\begin{array}{lll} -1 & 1 & 0\\ 0 & -1 & 0\\ 1 & 0 & -1 \end{array}\right),\;\tilde{X}={\left(\delta_{ai}^{T},v_{ai}^{T},\Omega^{T},{\dot{\Omega }}^{T},\delta_{gi}^{T},v_{gi}^{T}\right)}^{T},I$ is the identity matrix, and ⊗ is the Kronecker product.Let $U=\left(\begin{array}{cc} U_{11} & U_{12}\\ U_{21} & U_{22} \end{array}\right)$=$\left(\begin{array}{llllll} k_{1}L_{s} & -k_{2}I_{l} & -k_{3}I_{l} & -k_{4}I_{l} & 0 & 0\\ 0 & 0 & 0 & 0 & g_{1}L_{s} & -g_{2}I_{k} \end{array}\right)$Substituting $\;U=\left(\begin{array}{cc} U_{11} & U_{12}\\ U_{21} & U_{22} \end{array}\right)\cdot \tilde{X}$ into Equation (19): $\dot{\tilde{X}}=-T_{l}\cdot \tilde{X}$where
$$T_{l}=\left[\begin{array}{llllll} 0 & I & 0 & 0 & 0 & 0\\ 0 & 0 & I & 0 & 0 & 0\\ 0 & 0 & 0 & I & 0 & 0\\ k_{1}L_{s} & -k_{2}I & -k_{3}I & -k_{4}I & 0 & 0\\ 0 & 0 & 0 & 0 & 0 & I\\ 0 & 0 & 0 & 0 & g_{1}L_{s} & -g_{2}I \end{array}\right]$$Elementary row and column transformation can be taken on $T_{l}$:$$T_{l}=\left[\begin{array}{llllll} I & 0 & 0 & 0 & 0 & 0\\ 0 & I & 0 & 0 & 0 & 0\\ 0 & 0 & 0 & I & 0 & 0\\ 0 & 0 & 0 & 0 & I & 0\\ 0 & 0 & k_{1}L_{s} & 0 & 0 & 0\\ 0 & 0 & 0 & 0 & 0 & g_{1}L_{s} \end{array}\right]=\left(\begin{array}{ccc} I_{1} & 0 & 0\\ 0 & 0 & I_{2}\\ 0 & k_{1}L & g_{1}L \end{array}\right)=E$$If $\mathrm{Rank}\left(L\right)=N-1,\mathrm{Rank}\left(\left[\begin{array}{cc} I_{1} & \\ & I_{2} \end{array}\right]\right)=r$, it can be seen that:$\;\mathrm{Rank}\left(T_{l}\right)=\mathrm{Rank}\left(E\right)=r+N-1$.where $T_{l}$ has only zero eigenvalues. Therefore, we must select the parameters $k_{1}$,$k_{2},k_{3},k_{4},g_{1},\mathrm{and}\;g_{2}$ so that $T_{l}$ has zero eigenvalue, and all other eigenvalues have negative real parts. The parameters $k_{1}$,$k_{2},k_{3}\;\mathrm{and}\;k_{4}$ need to meet the stability of UAV consistency, and the parameters $g_{1}$ and $g_{2}$ need to meet the stability of UGV consistency. After determining the parameters, $T_{l}$ can be converted to a Jordan standard type: $T_{l}=PJP^{-1}$. Let $v_{1}^{T}$ be the first row of $P^{-1}$ and the left eigenvector have eigenvalue 0. Let $w_{1}$ be the first column of P and the right eigenvector with eigenvalue 0. Therefore, $v_{1}^{T}w_{1}=1$; when the time approaches infinity, the system’s state becomes: $\lim_{t\to \infty }^{\;}\tilde{X}=\lim_{t\to \infty }^{\;}e^{T_{l}t}\tilde{X}\left(0\right)$, $e^{T_{l}t}\tilde{X}\left(0\right)\to \left(w_{1}v_{1}^{T}\right)\tilde{X}\left(0\right)\left(t\to \infty \right)$. According to Lemma 1 it is then seen that the system can reach asymptotic consensus in cases where time tends toward infinity. □ 

### 3.4. Heterogeneous Cooperative Formation Control

For UAV:



(21)
$${\tilde{u}}_{ia}=\alpha \sum_{j\in N_{i}}^{\;}a_{ij}\left(\delta_{j}-\delta_{i}\right)-\beta a_{ij}\left(v_{j}-v_{i}\right)-\gamma_{1}\Omega_{i}-\gamma_{2}{\dot{\Omega }}_{i}$$



For UGV:



(22)
$${\tilde{u}}_{ig}=\alpha \sum_{j\in N_{i}}^{\;}a_{ij}\left(\delta_{j}-\delta_{i}\right)-\beta a_{ij}\left(v_{j}-v_{i}\right)$$


$$\delta_{i}={\left(\delta_{ia}^{T},\delta_{ig}^{T}\right)}^{T},\;v_{i}={\left(v_{ia}^{T},v_{ig}^{T}\right)}^{T},{\;\Omega }_{i}={\left(g\theta ,-g\phi ,0\right)}^{T},\dot{\Omega }={\left(g\dot{\theta },-g\dot{\phi },0\right)}^{T}$$



In combination with the Laplace matrix, Protocols (21) and (22) are rewritten as:(23)$$\tilde{U}=H\cdot \tilde{X}$$
$$H=\left(\begin{array}{cc} \begin{array}{l} \alpha L_{AA}⊗I_{3}\\ \alpha L_{GA}⊗I_{3} \end{array} & \begin{array}{cc} \begin{array}{l} -\beta L_{AA}⊗I_{3}\\ -\beta L_{GA}⊗I_{3} \end{array} & \begin{array}{cc} \begin{array}{l} -\gamma_{1}I_{l}⊗I_{3}\\ 0 \end{array} & \begin{array}{cc} \begin{array}{l} {-\gamma }_{2}I_{l}⊗I_{3}\\ 0 \end{array} & \begin{array}{cc} \begin{array}{l} \alpha L_{AG}⊗I_{3}\\ \alpha L_{GG}⊗I_{3} \end{array} & \begin{array}{l} -\beta L_{AG}⊗I_{3}\\ -\beta L_{GG}⊗I_{3} \end{array} \end{array} \end{array} \end{array} \end{array} \end{array}\right)$$

Define the state-space form of the heterogeneous multi-agent formation:(24)$$\dot{\tilde{X}}=A\tilde{X}+B\tilde{U}$$
where $A=\left(\begin{array}{ll} A_{A} & 0\\ 0 & A_{G} \end{array}\right),\;B=\left(\begin{array}{cc} B_{A} & 0\\ 0 & B_{G} \end{array}\right),\tilde{X}={\left({\overset{¯}{X}}^{T},{\overset{¯}{W}}^{T}\right)}^{T}.$
**Theorem** **3.***if Protocols (21) and (22) meet*$\alpha >0,\beta >0,\gamma_{1}>0,\gamma_{2}>0,\beta \gg \alpha ,\gamma_{1}>\beta$*,*$\gamma_{2}>\beta ,\;\beta \gamma_{1}\gamma_{2}>\beta^{2}\gamma_{2}\alpha$*, the heterogeneous system in (7) can be achieved, and (8) is defined in the formation, then formation control is realized.*
**Proof of Theorem** **3.**Substitute Equation (23) into Equation (24) and obtain: $\dot{\tilde{X}}=-T_{S}\cdot \tilde{X}$where $T_{s}=\left[\begin{array}{cccccc} 0 & 1 & 0 & 0 & 0 & 0\\ 0 & 0 & 1 & 0 & 0 & 0\\ 0 & 0 & 0 & 1 & 0 & 0\\ \alpha L_{AA}⊗I_{3} & -\beta L_{AA}⊗I_{3} & -\gamma_{1}I_{l}⊗I_{3} & -\gamma_{2}I_{l}⊗I_{3} & \alpha L_{AG}⊗I_{3} & -\beta L_{AG}⊗I_{3}\\ 0 & 0 & 0 & 0 & 0 & 1\\ \alpha L_{GA}⊗I_{3} & -\beta L_{GA}⊗I_{3} & 0 & 0 & \alpha L_{GG}⊗I_{3} & -\beta L_{GG}⊗I_{3} \end{array}\right],$I is the identity matrix and ⊗ is the Kronecker product.Internal stability parameters $\gamma_{1}$ and $\gamma_{2}$ should guarantee the stability of a single UAV agent. Therefore, the UAV is written as:$\left(\begin{array}{c} {\dot{P}}_{A}\\ {\dot{V}}_{A}\\ {\dot{\Omega }}_{A}\\ {\ddot{\Omega }}_{A} \end{array}\right)=\Gamma \left(\begin{array}{c} P_{A}\\ V_{A}\\ \Omega_{A}\\ {\dot{\Omega }}_{A} \end{array}\right)$, where $\Gamma =\left(\begin{array}{cccc} 0 & I & 0 & 0\\ 0 & 0 & I & 0\\ 0 & 0 & 0 & I\\ -\alpha I & -\beta I & -\gamma_{1}I & -\gamma_{2}I \end{array}\right)$.The characteristic polynomial is: $\det\left(SI-\Gamma \right)=\left|\begin{array}{cccc} S & -I & 0 & 0\\ 0 & S & -I & 0\\ 0 & 0 & S & -I\\ \alpha I & \beta I & \gamma_{1}I & S+\gamma_{2}I \end{array}\right|$where I is the identity matrix. For a single UAV:$$\left|SI-\Gamma \right|=s^{4}+\gamma_{2}s^{3}+\gamma_{1}s^{2}+\beta s+\alpha$$According to the Routh–Hurwitz stability criterion:$$\begin{array}{c} \alpha >0,\beta >0,\gamma_{1}>0,\gamma_{2}>0,\gamma_{1}\gamma_{2}>\beta ,\gamma_{1}\gamma_{2}\beta >\beta^{2}+\gamma_{2}^{2} \end{array}$$According to the linear stability theorem, the parameter $\alpha ,\beta ,\gamma_{1}$, and $\gamma_{2}$ should be selected so that there is a zero eigenvalue and other eigenvalues have negative genuine part parameters. The parameters α and β must meet the stability of UGV consistency, and the parameters $\gamma_{1}$ and $\gamma_{2}$ must meet the stability of UAV consistency. After selecting the parameters, $T_{s}$ can be converted to a Jordan standard type: $T_{s}=PJP^{-1}$. Let $v_{1}^{T}$ be the first row of $P^{-1}$ and the left eigenvector have eigenvalue 0. Let $w_{1}\;\mathrm{be}$ the first column of P and the right eigenvector have eigenvalue 0. Therefore, $v_{1}^{T}w_{1}=1$; when the time approaches infinity, the system’s state becomes: $\underset{t\to \infty }{\lim}\tilde{X}=\underset{t\to \infty }{\lim}e^{T_{s}t}\tilde{X}\left(0\right)$, $e^{T_{s}t}\tilde{X}\left(0\right)\to \left(w_{1}v_{1}^{T}\right)\tilde{X}\left(0\right)\left(t\to \infty \right)$. According to Lemma 1, when the time approaches infinity, the system in (7) asymptotically agrees; that is, the system achieves cooperative formation.□ 

### 3.5. Heterogeneous Cooperative Optimal Formation Control

UAV systems and UGV systems are heterogeneous systems. Applying the optimal control law to the heterogeneous system can be expressed as:



(25)
$${\tilde{u}}_{ia}=k_{1}\sum_{j\in N_{i}}^{\;}a_{ij}\left(\delta_{j}-\delta_{i}\right)-k_{2}a_{ij}\left(v_{j}-v_{i}\right)-k_{3}\Omega_{i}-k_{4}{\dot{\Omega }}_{i}$$



(26)$${\tilde{u}}_{ig}=g_{1}\sum_{j\in N_{i}}^{\;}a_{ij}\left(\delta_{j}-\delta_{i}\right)-g_{2}a_{ij}\left(v_{j}-v_{i}\right)$$ where $k_{1},k_{2},k_{3}$,and $k_{4}$ are derived from the matrix $K.g_{1}$ and $g_{2}$ are derived from the matrix G.

In combination with the Laplace matrix, Protocols (25) and (26) are rewritten as:(27)$$\tilde{U}=S\cdot \tilde{X}$$
$$S=\left(\begin{array}{llllll} k_{1}L_{AA}⊗I_{3} & -k_{2}L_{AA}⊗I_{3} & -k_{3}I_{l}⊗I_{3} & -k_{4}I_{l}⊗I_{3} & k_{1}L_{AG}⊗I_{3} & -k_{2}L_{AG}⊗I_{3}\\ g_{1}L_{GA}⊗I_{3} & -g_{2}L_{GA}⊗I_{3} & 0 & 0 & g_{1}L_{GG}⊗I_{3} & -g_{2}L_{GG}⊗I_{3} \end{array}\right)$$

Define the state space of heterogeneous multi-agent formation:(28)$$\dot{\tilde{X}}=A\tilde{X}+B\tilde{U}$$
where $A=\left(\begin{array}{cc} A_{A} & 0\\ 0 & A_{G} \end{array}\right),\;B=\left(\begin{array}{cc} B_{A} & 0\\ 0 & B_{G} \end{array}\right),\;\tilde{X}={\left({\overset{¯}{X}}^{T},{\overset{¯}{W}}^{T}\right)}^{T}$
**Theorem** **4.***Heterogeneous systems can complete formation and optimize Performance Functions (13) and (14) if they use Protocols (25) and (26), respectively.*
**Proof of Theorem** **4.**Each UAV and UGV is defined as a group of formation units, so their number is made the same, that is, L = K; the block Laplacian matrix of the system has the same number of rows and columns. Then, the heterogeneous system in (7) becomes:(29)$$\dot{X}=\hat{A}X+\hat{B}U$$
where $\hat{A}=\left(\begin{array}{cccccc} 0 & I & 0 & 0 & 0 & 0\\ 0 & 0 & 0 & 0 & 0 & 0\\ 0 & 0 & 0 & I & 0 & 0\\ 0 & 0 & 0 & 0 & I & 0\\ 0 & 0 & 0 & 0 & 0 & I\\ 0 & 0 & 0 & 0 & 0 & 0 \end{array}\right)$,$\hat{B}=\left(\begin{array}{c} 0----0\\ I----0\\ 0----0\\ 0----0\\ 0----0\\ 0----I \end{array}\right)$.According to State Equation (29):$$\left(\begin{array}{c} SI-\hat{A}----B \end{array}\right)=\left(\begin{array}{cccccccc} S & -I & 0 & 0 & 0 & 0 & 0 & 0\\ 0 & S & 0 & 0 & 0 & 0 & I & 0\\ 0 & 0 & S & -I & 0 & 0 & 0 & 0\\ 0 & 0 & 0 & S & -I & 0 & 0 & 0\\ 0 & 0 & 0 & 0 & S & -I & 0 & 0\\ 0 & 0 & 0 & 0 & 0 & S & 0 & I \end{array}\right)$$It is easy to discover the eigenvalues of the matrix $\hat{A}$: $\lambda_{1}=\lambda_{2}=\cdots =\lambda_{6}=0$.Verify the rank of the matrix $\left(\begin{array}{c} SI-\hat{A}----B \end{array}\right)$ with the above eigenvalues. Let $S=\lambda_{1}=\lambda_{2}=\ldots =\lambda_{6}=0$:Rank $\left(\begin{array}{c} SI-\hat{A}----B \end{array}\right)\;$ = Rank $\left(\begin{array}{cccccccc} 0 & -I & 0 & 0 & 0 & 0 & 0 & 0\\ 0 & 0 & 0 & 0 & 0 & 0 & I & 0\\ 0 & 0 & 0 & -I & 0 & 0 & 0 & 0\\ 0 & 0 & 0 & 0 & -I & 0 & 0 & 0\\ 0 & 0 & 0 & 0 & 0 & -I & 0 & 0\\ 0 & 0 & 0 & 0 & 0 & 0 & 0 & I \end{array}\right)\;$= 6, according to the rank criterion of PBH, the system state space is controllable.Let $\tilde{U}=\left(\begin{array}{cc} U_{11} & U_{12}\\ U_{21} & U_{22} \end{array}\right)$=$\left(\begin{array}{llllll} k_{1}L_{AA} & -k_{2}L_{AA} & -k_{3}I & -k_{4}I & k_{1}L_{AG} & -k_{2}L_{AG}\\ g_{1}L_{GA} & -g_{2}L_{GA} & 0 & 0 & g_{1}L_{GG} & -g_{2}L_{GG} \end{array}\right)$Substituting $\;\tilde{U}=\left(\begin{array}{cc} U_{11} & U_{12}\\ U_{21} & U_{22} \end{array}\right)\cdot \tilde{X}$ into Equation (28):$\;\dot{\tilde{X}}=-T*\cdot \tilde{X}$where
$$T*=\left[\begin{array}{llllll} 0 & I & 0 & 0 & 0 & 0\\ 0 & 0 & I & 0 & 0 & 0\\ 0 & 0 & 0 & I & 0 & 0\\ k_{1}L_{AA} & -k_{2}L_{AA} & -k_{3}I & -k_{4}I & g_{1}L_{AG} & -g_{2}L_{AG}\\ 0 & 0 & 0 & 0 & 0 & I\\ k_{1}L_{GA} & -k_{2}L_{GA} & 0 & 0 & g_{1}L_{GG} & -g_{2}L_{GG} \end{array}\right]$$The T* basic determinant change:$$\left(\begin{array}{cccccc} \lambda +I & -I & 0 & 0 & 0 & 0\\ I & \lambda & -I & 0 & 0 & 0\\ 2I & 0 & \lambda -I & -I & 0 & 0\\ C_{1} & C_{2} & k_{3}I & \lambda +k_{4}I & -g_{1}L_{AG} & g_{2}L_{AG}\\ I & 0 & 0 & 0 & \lambda & -I\\ C_{3} & k_{2}L_{GA}-I & 0 & 0 & -g_{1}L_{GG} & g_{2}L_{GG} \end{array}\right)=\lambda I-\Lambda ,$$
where
$$\Lambda =\left(\begin{array}{cccccc} -I & I & 0 & 0 & 0 & 0\\ -I & 0 & I & 0 & 0 & 0\\ -2I & 0 & I & I & 0 & 0\\ -C_{1} & -C_{2} & -k_{3}I & -k_{4}I & g_{1}L_{AG} & -g_{2}L_{AG}\\ -I & 0 & 0 & 0 & 0 & I\\ -C_{3} & I-k_{2}L_{GA} & 0 & 0 & g_{1}L_{GG} & -g_{2}L_{GG} \end{array}\right)$$
$$C_{1}=I-k_{1}L_{AA}-g_{2}L_{AG}-k_{2}L_{AA}-k_{4}I.$$
$$C_{2}=k_{2}L_{AA}-k_{3}I-I$$
$$C_{3}=I-k_{1}L_{GA}-g_{2}L_{GG}$$Now λI−T*≅λI−Λ, so matrix T* is similar to Λ. Therefore, there is a nonsingular transformation matrix Q, making $\Lambda =QT*Q^{-1}$,$\dot{\tilde{X}}=\Lambda \cdot \tilde{X};\;\Lambda$ is the matrix for which the sum of each row is zero. Therefore, there is at least one zero eigenvalue. Elementary row and column transformation can be taken on T*:$$T*=\left(\begin{array}{cccccc} I & 0 & 0 & 0 & 0 & 0\\ 0 & I & 0 & 0 & 0 & 0\\ 0 & 0 & 0 & 0 & I & 0\\ 0 & 0 & 0 & 0 & 0 & I\\ 0 & 0 & k_{1}L_{AA} & -k_{2}L_{AA} & g_{1}L_{AG} & -g_{2}L_{AG}\\ 0 & 0 & k_{1}L_{GA} & -k_{2}L_{GA} & g_{1}L_{GG} & -g_{2}L_{GG} \end{array}\right)=\left(\begin{array}{ccc} I_{1} & 0 & 0\\ 0 & 0 & I_{2}\\ 0 & kL & gL \end{array}\right)=E$$If Rank(L)=N−1,$\mathrm{Rank}\left(\left[\begin{array}{cc} I_{1} & \\ & I_{2} \end{array}\right]\right)=r$, it can be seen that:$$\mathrm{Rank}\left(T^{*}\right)=\mathrm{Rank}\left(\Lambda \right)=\mathrm{Rank}\left(E\right)\;=r\;+N-1$$
where the Λ and T* have only zero eigenvalues. Therefore, we must select the parameters $k_{1}$,$k_{2},k_{3},k_{4},g_{1},\mathrm{and}\;g_{2}$ so that T* has zero eigenvalue and all other eigenvalues have negative real parts. The parameters $k_{1}$,$k_{2},k_{3},\;\mathrm{and}\;k_{4}$ must meet the stability of UAV consistency, and the parameters $g_{1}$ and $g_{2}$ must meet the stability of UGV consistency. After determining the parameters, T* can be converted to a Jordan standard type: $T*=PJP^{-1}$. Let $v_{1}^{T}$ be the first row of $P^{-1}$ and the left eigenvector have eigenvalue 0. Let $w_{1}$ the first column of P and the right eigenvector have eigenvalue 0. Therefore, $v_{1}^{T}w_{1}=1$; when the time approaches infinity, the system’s state becomes: $\underset{t\to \infty }{\lim}\tilde{X}=\underset{t\to \infty }{\lim}e^{T^{*}t}\tilde{X}\left(0\right)$, $e^{T^{*}t}\tilde{X}\left(0\right)\to \left(w_{1}v_{1}^{T}\right)\tilde{X}\left(0\right)\left(t\to \infty \right)$. According to Lemma 1, it is then seen that the system can reach asymptotic consensus in cases where time tends toward infinity.□ 


## 4. Simulations

The formation protocol, distributed optimal formation protocol, cooperative formation protocol, and cooperative optimal formation protocol individually designed in this paper are simulated and analyzed using Matlab2016a. The effectiveness of the designed control protocol is verified via simulation. To achieve a better cooperative formation task effect in the system, the speed values set by the UAV system and the UGV system are similar.

The initial state is as follows: the position of UAV1 is (30,50,50)m, the position of UAV2 is (90,30,50)m, and the position of UAV3 is (60,30,15)m. The speed of UAV1 is (1,1,−2)m/s, the speed of UAV2 is (1,−2,1)m/s, and the speed of UAV3 is (−2,1,1)m/s. The attitude angle of UAV1 is ${\left(0,0,0\right)}^{\circ }$, the attitude angle of UAV2 is ${\left(0,0,0\right)}^{\circ }$, and the attitude angle of UAV3 is ${\left(0,0,0\right)}^{\circ }$

The rate of the attitude angle of UAV1 is ${\left(0,0,0\right)}^{\circ }/s$, the rate of the attitude angle of UAV2 is ${\left(0,0,0\right)}^{\circ }/s$, and the rate of the attitude angle of UAV3 is ${\left(0,0,0\right)}^{\circ }/s$. The setting position of UAV1 is (10,10,30)m, the setting position of UAV2 is (0,5,30)m, and the setting position of UAV3 is (0,15,30)m. The UAV and UGV systems have a set speed of (1,0,0)m/s. The position of UGV1 is (90,50,0)m, the position of UGV2 is (65,10,0)m, and the position of UGV3 is (30,20,0)m. The speed of UGV1 is (1,1,0)m/s, the speed of UGV2 is (2,−2,0)m/s, and the speed of UGV3 is (−1,1,0)m/s. The setting position of UGV1 is (10,10,0)m, the setting position of UGV2 is (0,5,0)m, and the setting position of UGV3 is (0,15,0)m.

The parameters are $\alpha =0.2,\beta =1.5,\gamma_{1}=5,\gamma_{2}=2,k_{1}=2.3452,k_{2}=6.4707,k_{3}=7.7541,k_{4}=4.5835,g_{1}=0.4472,\mathrm{and}\;g_{2}=1.0461$.

Simulations of the UAV system and the UGV system using formation control and distributed optimal formation control are shown in Figure 1 and Figure 2. By comparing the two figures in Figure 1 and Figure 2, it can be found that both protocols can be used to complete triangular formation at the same time, but when distributed optimal formation control is used, it can be observed that the error between the actual position and the set value is significantly reduced in a short period of time.

When using the formation control protocol, with parameters $\alpha =0.2,\beta =1.5,\gamma_{1}=5,\mathrm{and}\;\gamma_{2}=2$. Figure 1 shows the change of position coordinates of the multi-agent system and shows the actual formation position of the UAV and UGV when t=10 s. It can be seen that there is a significant error with the set value.

When using the distributed optimal formation control protocol, with parameters $k_{1}=2.3452,k_{2}=6.4707,k_{3}=7.7541,\;k_{4}=4.5835,g_{1}=0.4472,\mathrm{and}\;g_{2}=1.0461$. Figure 2 shows the position coordinate changes of the system and shows the actual formation status of the UAV and UGV when t=10 s. It can be observed that the system can quickly complete the triangular formation and that the error between the actual position and the set value is small.

It can be seen from Figure 3 and Figure 4 that the state of each system variable changes with time when the formation control protocol is used. It can be seen from the figures that the state of each variable in the system is stable from 40 s to 50 s.

From Figure 5 and Figure 6, it can be seen that the system uses the distributed optimal formation control protocol to change the state of each variable with time. It is observed that the system reaches stability between 30 and 40 s.

The simulation of cooperative formation protocol and cooperative optimal formation protocol of heterogeneous multi-agents is shown in Figure 7 and Figure 8.

When using the cooperative formation control protocol, with parameters $\alpha =0.2,\beta =1.5,\gamma_{1}=5,\mathrm{and}\;\gamma_{2}=2$. Figure 7 shows the change of position coordinates of the system and the expected formation state of the system at the 10th second. It can be seen that the actual formation position of the system has a significant error with the set value.

When using the cooperative optimal formation control protocol, with parameter $k_{1}=2.3452,k_{2}=6.4707,k_{3}=7.7541,k_{4}=4.5835,g_{1}=0.4472,\mathrm{and}\;g_{2}=1.0461$.

Figure 8 shows the change in position coordinates of the system and the expected formation state of the system at t=10 s. It can be observed that the system can quickly complete the triangular formation and that the error between the actual position and the set value is small. In addition, the cooperative optimal formation protocol speeds up the convergence rate of the system, which is of great help to the formation time of the system.

When using the formation control and the distributed optimal formation control, the formation states of the UAV system at different times are shown in Figure 9 and Figure 10.

Figure 9 shows the formation status of the UAV system at different times under the use of the formation control protocol. It can be observed that the system can complete the triangle formation but also that there is a relative reach error with the set position in the formation completion process and that there is still an error at the fiftieth second.

Figure 10 shows the formation status of the UAV system at different times under the use of the distributed optimal formation control protocol. It can be observed that the system can quickly complete the triangle formation and that the error between the actual position and the set value is very small at the thirtieth second.

When using the formation control and the distributed optimal formation control, the formation states of the UGV system at different times are shown in Figure 11 and Figure 12. 

Figure 11 shows the formation status of the UGV system at different moments when the formation control protocol is used. It can be observed that the UGV system can complete the triangle formation, but there is still a certain error with the set value in the formation completion process. At the thirtieth second, the difference between the actual position and the set value gradually decreases. 

Figure 12 shows the change in the formation shape of the UGV system at different moments when the distributed optimal formation control protocol is used; it can be observed that the UGV system can quickly complete the triangle formation and that the error between the actual position and the set value is small.

When using the heterogeneous cooperative formation control and the heterogeneous cooperative optimal formation control, the formation states of the UAV system at different times are shown in Figure 13 and Figure 14.

Figure 13 shows the formation status of the UAV system at different times under the use of the heterogeneous cooperative formation control protocol. It is observed that the system can complete the triangle formation. At the thirtieth second, the difference between the actual position and the set value gradually decreases. 

Figure 14 shows the formation status of the UAV system at different times under the use of the heterogeneous cooperative optimal formation control. It can be observed that the system can quickly complete the triangle formation and that the error between the actual position and the set value is very small at the tenth second.

When using the heterogeneous cooperative formation control and the heterogeneous cooperative optimal formation control, the formation states of the UGV system at different times are shown in Figure 15 and Figure 16.

Figure 15 shows the formation status of the UGV system at different times under the use of the heterogeneous cooperative formation control protocol. It can be observed that the system can complete the triangle formation. At the thirtieth second, the difference between the actual position and the set value gradually decreases. 

Figure 16 shows the formation status of the UGV system at different times under the use of the heterogeneous cooperative optimal formation control. It can be observed that the system can quickly complete the triangle formation and that the error between the actual position and the set value is very small at the tenth second.

In order to fully verify the theoretical results, the structure is complicated, and the communication Laplacian matrix is as follows:$$L=\left(\begin{array}{ccccccccc} -3 & 1 & 1 & 0 & 0 & 1 & 0 & 0 & 0\\ 1 & -3 & 1 & 0 & 1 & 0 & 0 & 0 & 0\\ 1 & 1 & -3 & 0 & 0 & 1 & 0 & 0 & 0\\ 0 & 0 & 1 & -3 & 1 & 1 & 0 & 0 & 0\\ 0 & 1 & 0 & 1 & -3 & 1 & 0 & 0 & 0\\ 1 & 0 & 0 & 1 & 1 & -3 & 0 & 0 & 0\\ 0 & 0 & 0 & 0 & 0 & 1 & -3 & 1 & 1\\ 0 & 0 & 0 & 0 & 1 & 0 & 1 & -3 & 1\\ 0 & 0 & 0 & 1 & 0 & 0 & 1 & 1 & -3 \end{array}\right)$$

The added set of UAV states is as follows: the position of UAV4 is (40,50,10)m, the position of UAV5 is (60,30,20)m, and the position of UAV 6 is (30,20,10)m. The speed of UAV4 is (1,1,1)m/s, the speed of UAV5 is (2,2,1)m/s, and the speed of UAV6 is (3,1,1)m/s.The attitude angle of UAV4 is ${\left(0,0,0\right)}^{\circ }$, the attitude angle of UAV5$\;\mathrm{is}\;{\left(0,0,0\right)}^{\circ }$, and the attitude angle of UAV6 is ${\left(0,0,0\right)}^{\circ }$. The setting position of UAV4 is (10,10,10)m, the setting position of UAV5 is (0,5,10)m, and the setting position of UAV6 is (0,15,10)m. The UAV systems have a set speed of (1,0,0)m/s. The simulation results are shown in Figure 17, Figure 18, Figure 19 and Figure 20.

The experiments have verified the formation control and distributed optimization formation control, as shown in Figure 17 and Figure 18, respectively. By comparing Figure 17 and Figure 18, it can be found that both protocols can be used to complete triangular formation at the same time, but when distributed optimal formation control is used, it can be observed that the error between the actual position and the set value is significantly reduced in a short period of time.

When using the formation control protocol, with parameters $\alpha =0.2,\beta =1.5,\gamma_{1}=5,\gamma_{2}=2$, Figure 17 shows the change of position coordinates of the complex system and the expected formation state of the complex system. However, it can be seen that there is a significant error with the set value.

When using the distributed optimal formation control protocol, with parameters $k_{1}=2.3452,k_{2}=6.4707,k_{3}=7.7541,\;k_{4}=4.5835,g_{1}=0.4472,\mathrm{and}\;g_{2}=1.0461$. Figure 18 shows the change of position coordinates of the complex system and the expected formation state of the complex system. It can be observed that the complex system can quickly complete the triangular formation and that the error between the actual position and the set value is small.

The experiments have verified the cooperative formation control and the cooperative optimal formation control, as shown in Figure 19 and Figure 20, respectively.

When using the cooperative formation control protocol, with parameters $\alpha =0.2,\beta =1.5,\gamma_{1}=5,\mathrm{and}\;\gamma_{2}=2$, Figure 19 shows the change in position coordinates of the complex system and the expected formation state of the complex system. However, it can be seen that there is a significant error with the set value. 

When using the cooperative optimal formation control protocol, with parameters $k_{1}=2.3452,k_{2}=6.4707,k_{3}=7.7541,\;k_{4}=4.5835,g_{1}=0.4472,\mathrm{and}\;g_{2}=1.0461$, Figure 20 shows the change in position coordinates of the complex system and the expected formation state of the complex system. It can be observed that the complex system can quickly complete the triangular formation and that the error between the actual position and the set value is small. In addition, the cooperative optimal formation protocol speeds up the convergence rate of the complex system, which is of great help to the formation time of the complex system.

## 5. Conclusions

In this paper, a heterogeneous multi-agent system has been established by analyzing the dynamics model of the unmanned ground vehicle and the unmanned aerial vehicle. Firstly, the formation control protocol is proposed based on the communication topology of a multi-agent system. Then, according to the internal state of a single agent, the optimal control law of a single agent system is designed using the optimal control theory, and the optimal control law is introduced into the system to achieve the distributed optimal formation. Finally, based on the cooperative architecture of the heterogeneous multi-agent system, the cooperative formation design of the heterogeneous multi-agent system is carried out, and the optimal control theory is introduced into the heterogeneous multi-agent system to realize the optimal cooperative formation of the heterogeneous system. The stability of the system is further analyzed by graph theory. The communication topology of the multi-agent system does not interfere with the protocol and the protocol can optimize the performance function while the system completes the formation task. The simulation results show that the optimal control can accelerate the convergence speed of the system and greatly help the system to quickly reach the desired formation state. In the next step, we plan to investigate the anomaly detection and recognition problems under heterogeneous multi-agent cooperation architecture, and we plan to apply the theoretical research results in practice to engineering applications.

## Figures and Tables

**Figure 1 entropy-24-01440-f001:**
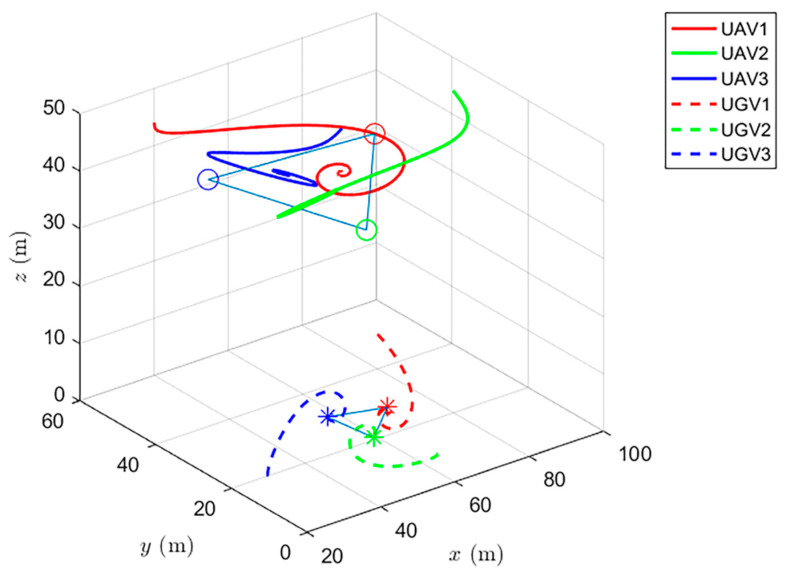
Formation control protocol for heterogeneous multi-agent systems.

**Figure 2 entropy-24-01440-f002:**
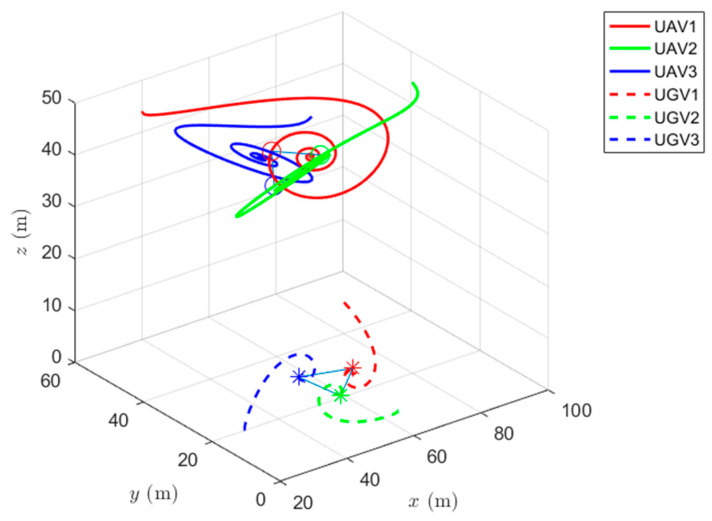
Distributed optimal formation control protocol for heterogeneous multi-agent systems.

**Figure 3 entropy-24-01440-f003:**
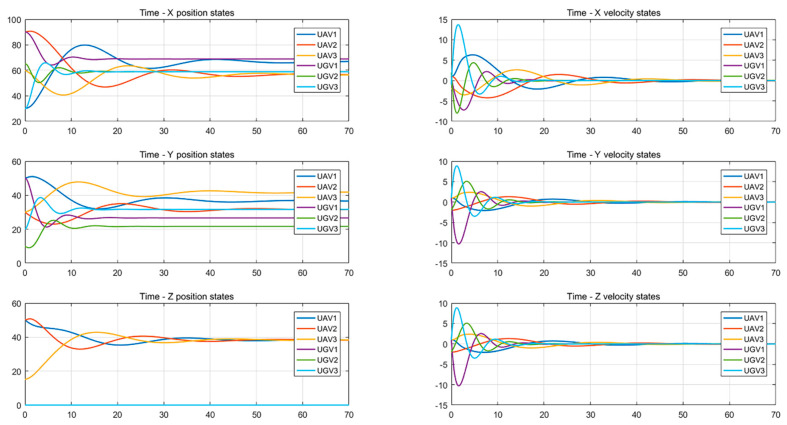
Position and velocity states change with time using the formation control protocol.

**Figure 4 entropy-24-01440-f004:**
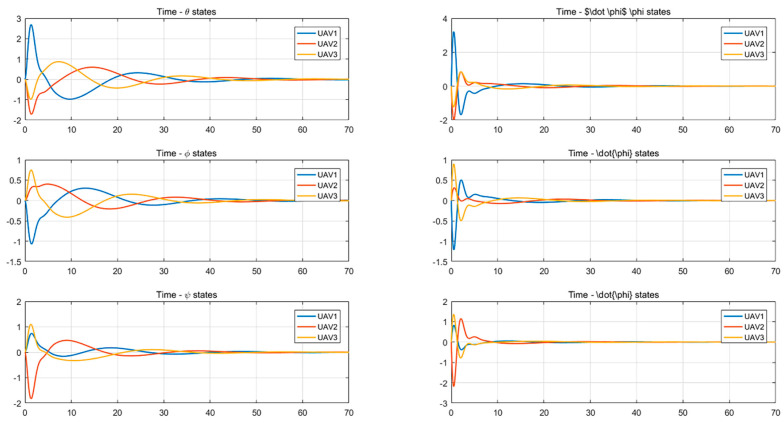
Attitude angle changes with time using a formation control protocol.

**Figure 5 entropy-24-01440-f005:**
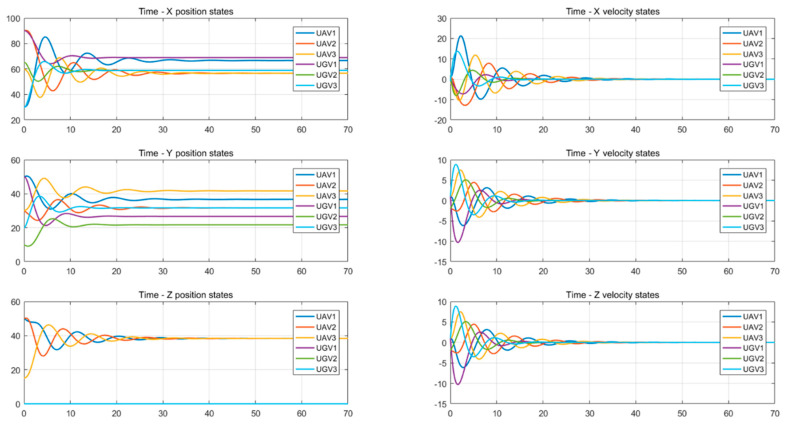
Changes in position and velocity states with time using the distributed optimal formation control protocol.

**Figure 6 entropy-24-01440-f006:**
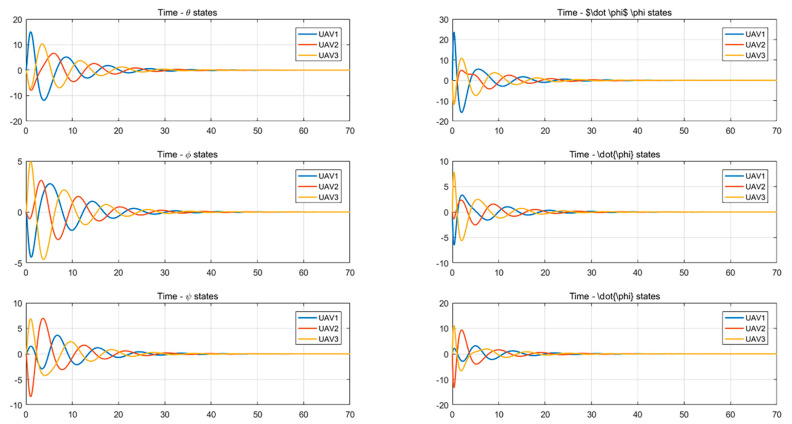
Attitude angle changes with time using the distributed optimal formation control protocol.

**Figure 7 entropy-24-01440-f007:**
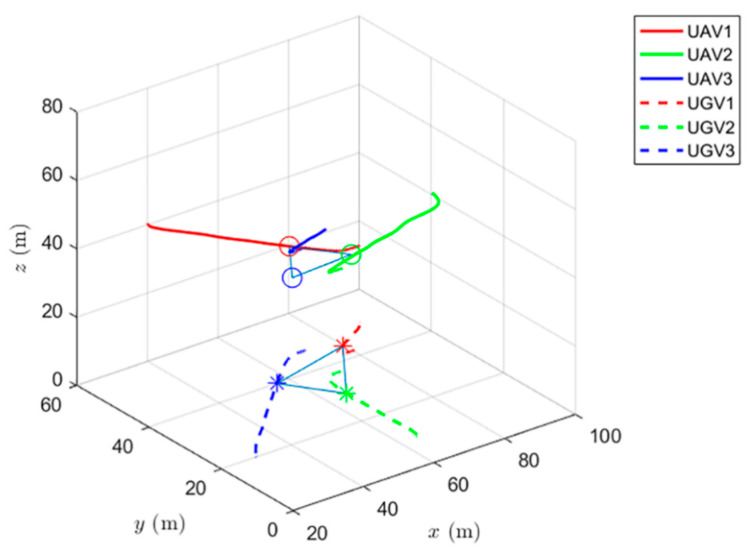
Distributed cooperative formation control for heterogeneous multi-agent systems.

**Figure 8 entropy-24-01440-f008:**
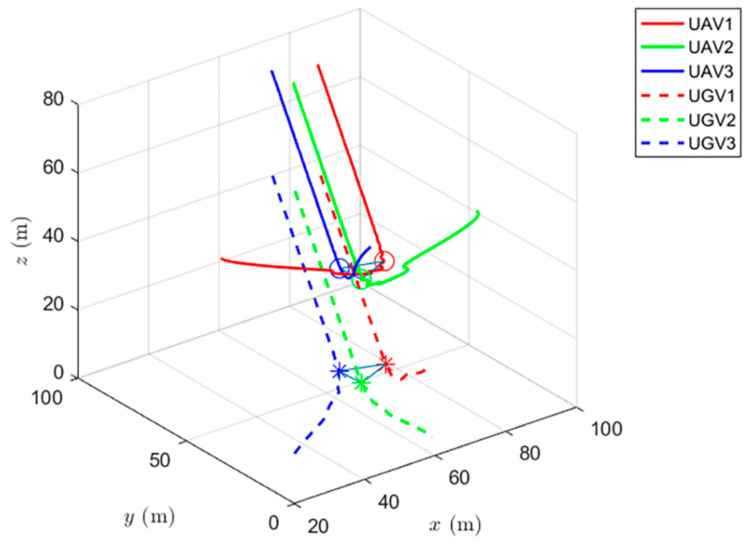
Distributed cooperative optimal formation control for heterogeneous multi-agent systems.

**Figure 9 entropy-24-01440-f009:**
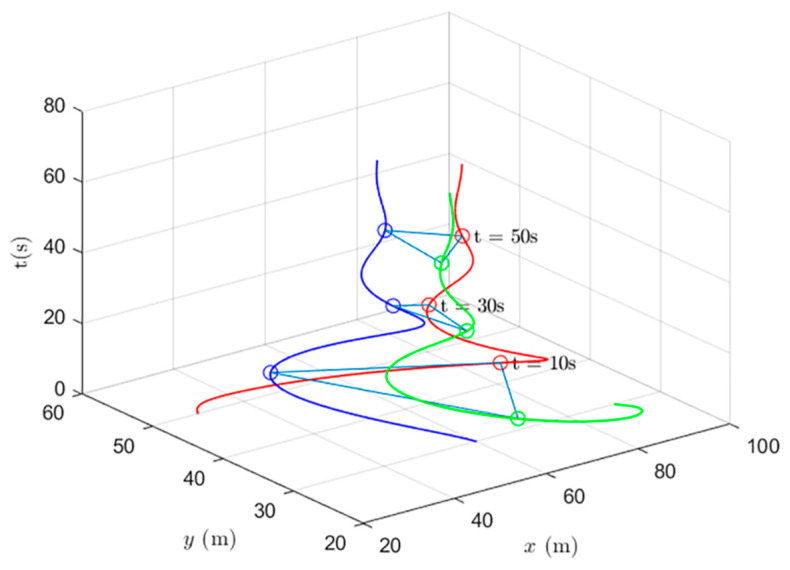
UAV system state formation based on formation control.

**Figure 10 entropy-24-01440-f010:**
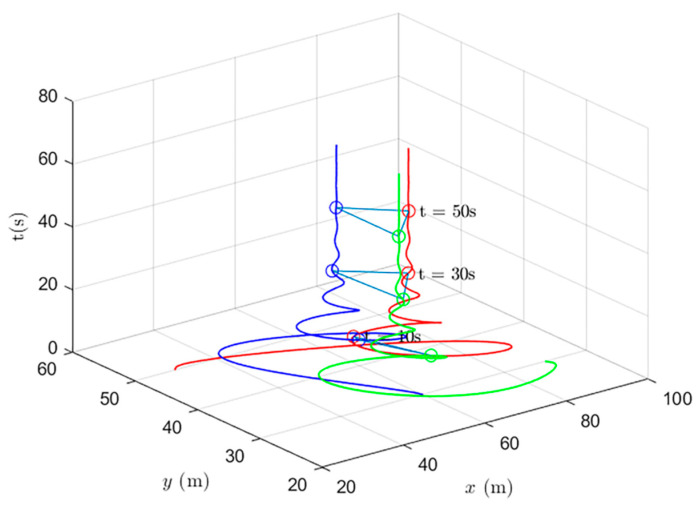
UAV system state formation based on distributed optimal formation control.

**Figure 11 entropy-24-01440-f011:**
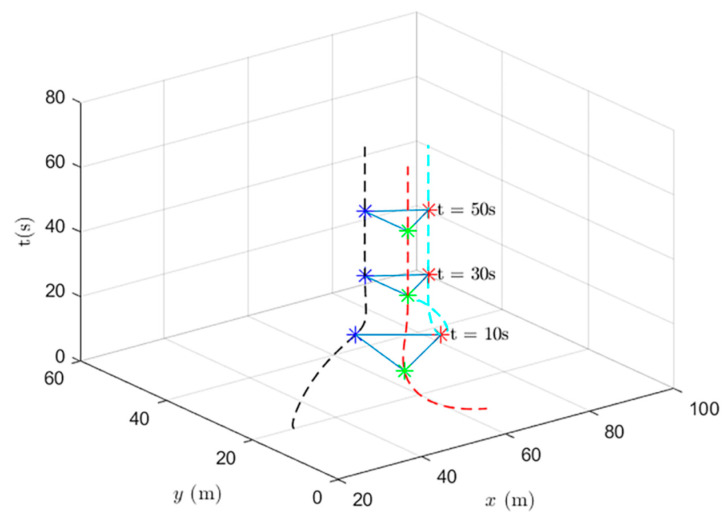
UGV system state formation based on formation control.

**Figure 12 entropy-24-01440-f012:**
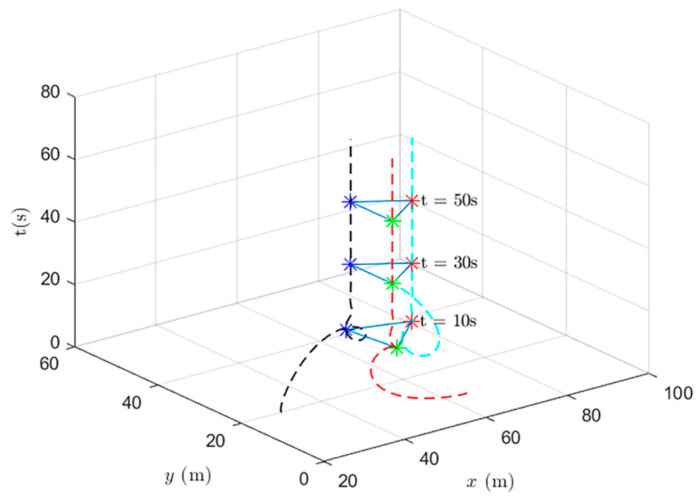
UGV system state formation based on distributed optimal formation control.

**Figure 13 entropy-24-01440-f013:**
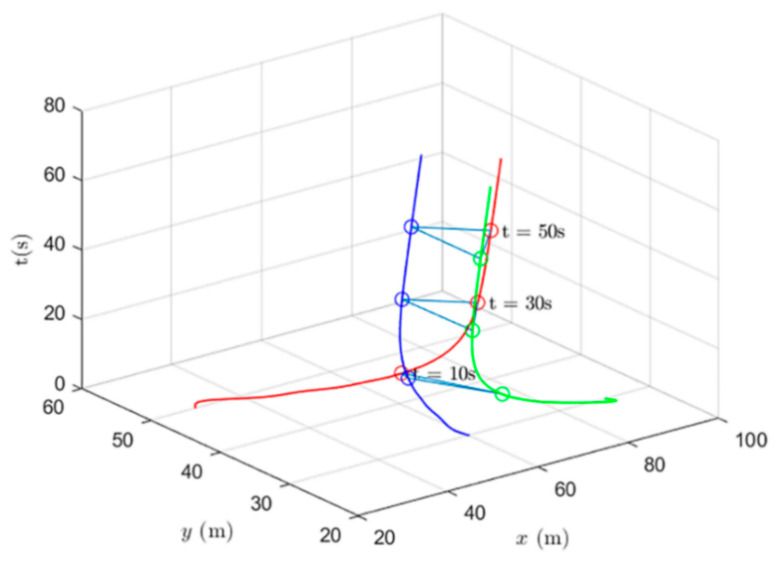
UAV system state formation based on heterogeneous cooperative formation control.

**Figure 14 entropy-24-01440-f014:**
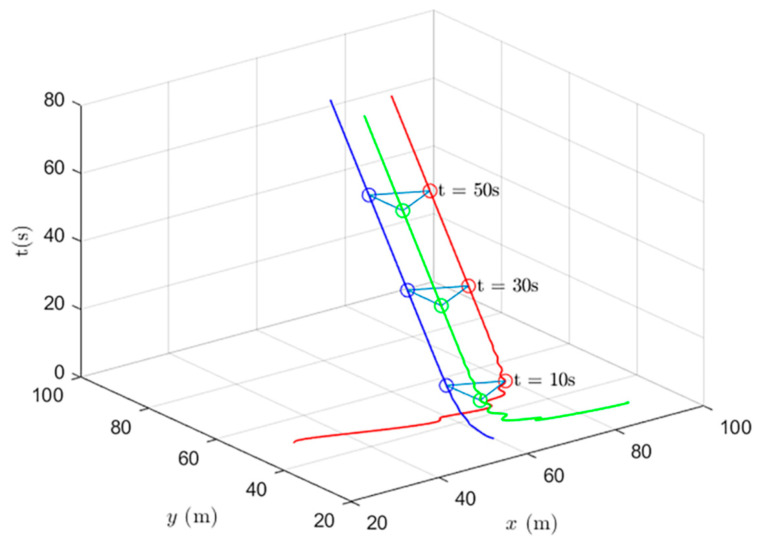
UAV system state formation based on heterogeneous cooperative optimal formation control.

**Figure 15 entropy-24-01440-f015:**
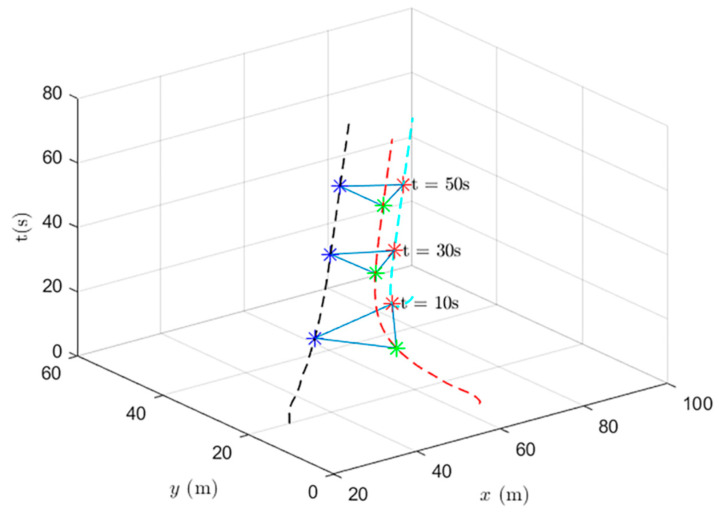
UGV system state formation based on heterogeneous cooperative formation control.

**Figure 16 entropy-24-01440-f016:**
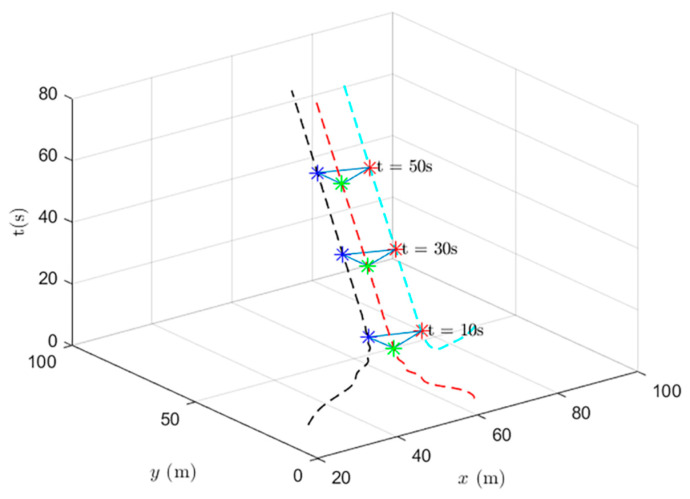
UGV system state formation based on heterogeneous cooperative optimal formation control.

**Figure 17 entropy-24-01440-f017:**
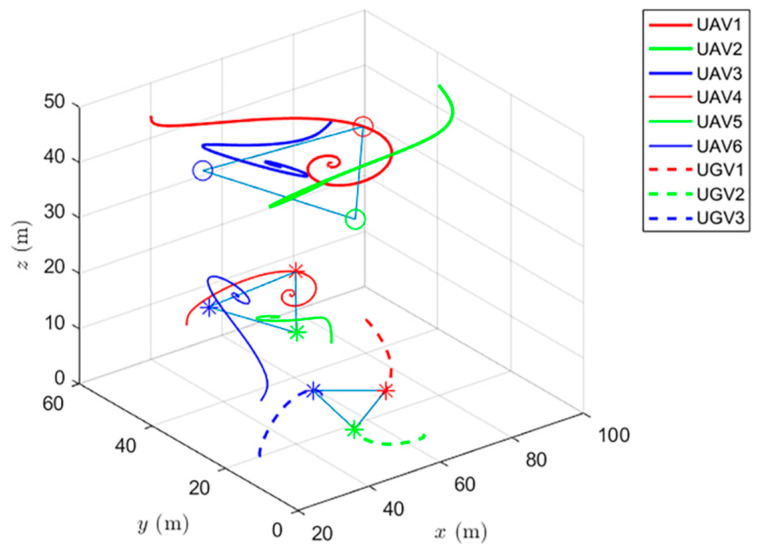
Formation control protocol for complex heterogeneous multi-agent systems.

**Figure 18 entropy-24-01440-f018:**
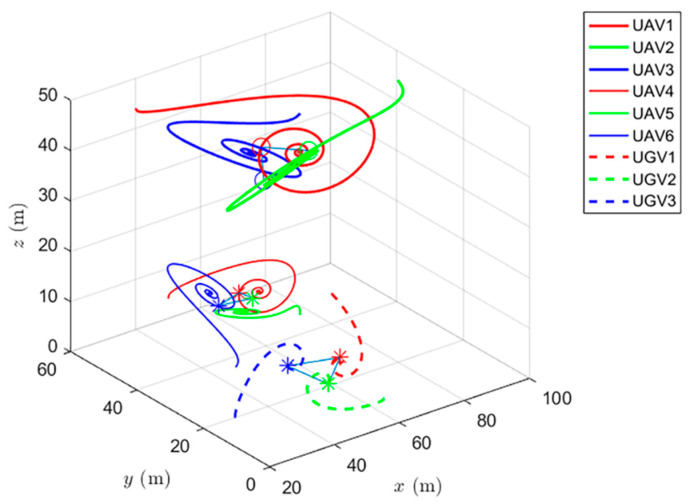
Distributed optimal formation control protocol for complex heterogeneous multi-agent systems.

**Figure 19 entropy-24-01440-f019:**
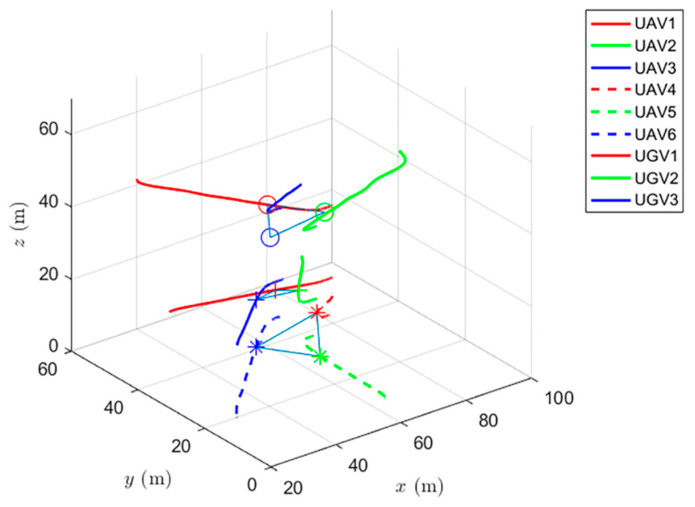
Distributed cooperative formation control for complex heterogeneous multi-agent systems.

**Figure 20 entropy-24-01440-f020:**
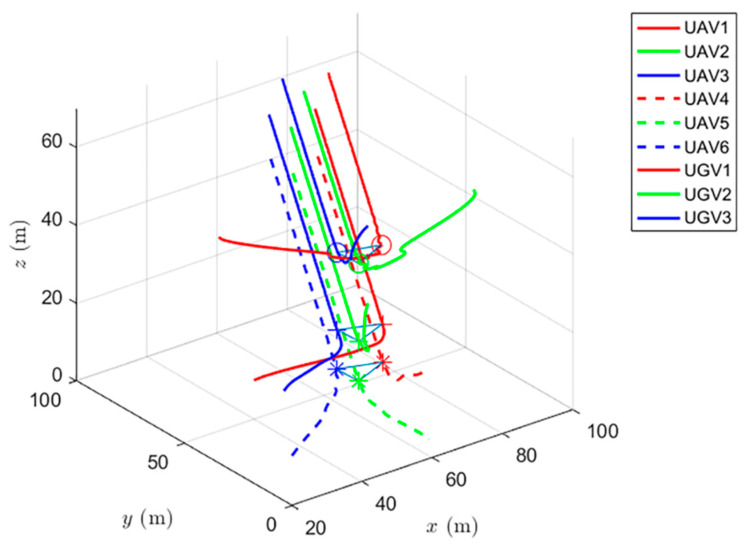
Distributed cooperative optimal formation control for complex heterogeneous multi-agent systems.

## Data Availability

The data that support the findings of this study are available from the corresponding author, Meichen Liu, upon reasonable request.

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
