# Peer review of "Collaborative Optimal Formation Control for Heterogeneous Multi-Agent Systems"

_entropy, 2022, doi:10.3390/e24101440_

Round 1

Reviewer 1 Report

The presentation of the paper can be improved. For example, the equation (2) and the matrices are very long. They should be better presented to improve understanding. Equation (5) needs improvement similarly. 

The proof of Theorem 2 seems to be questionable. The Jordan standard type is not always consistent with the diagonal form in the current setting. The inverse of P should be replaced with the generalized inverse and then the whole proof should be rewritten.

Theorem 4 is also written at a high level with the details hidden in the argument. I think the proofs should be reworked for most of the results although the results seem to be correct. It is very hard to follow the argument. 

The simulation is a weak point of this paper. It is of a very small scale. I suggest considering some larger networks and using real-world examples to validate the results. 

A further concern is regarding the relevance of this paper to the concept of 'entropy'. This should be justified to be published in this journal. 

Author Response

Thank the reviewers’comments concerning our manuscript entitled “Collaborative optimal formation control for heterogeneous multi-agent systems including UGV and UAV”(ID: entropy-1890420). Those comments are all valuable and very helpful for revising and improving our paper, as well as the important guiding significance to our researches.Special thanks to you for your good comments. I have made modifications in Word version author-coverletter-21857155.Please see the Word.

Reviewer 2 Report

After reading this manuscript, the reviewer recommends the following points:

1.     Abstract should overview the obtained results and Authors claims.

2.     In section 1 the authors tried to enumerate works without having connections between different references.

3.     What are the main differences between the current manuscript and the article:

4.     S. Liang, F. Y. Wang, Z. Q. Chen, and Z. X. Liu, "Formation control for discrete-time heterogeneous multi-agent systems,"426 INTERNATIONAL JOURNAL OF ROBUST AND NONLINEAR CONTROL, vol. 32, no. 10, pp. 5848-5865, JUL102022.

5.     I believe the method developed in this study is simply a combanition and application of existing designs. Similar ideas can also be found in the literature.

6.     The contribution of the manuscript is not clear to me compared to what is already in the literature. A more detailed discussion regarding the major contribution with regard to the newest existing works should be given to highlight the motivation of this work.

7.     The authors should clearly write the contribution of this paper.

8.     The detailed block diagram of the proposed approach should be added to clarify the design procedure and controller structure.

9.     It would be interesting if some comparisons with other literature methods could be included.

10.  More cases need to be examined to verify the robustness of the proposed methodology.

11.  The proposed method must be verified via experimental studies.

12.  Please provide clear advantages and limitations of the proposed methodology in the conclusions section.

Author Response

Thank the reviewers’comments concerning our manuscript entitled “Collaborative optimal formation control for heterogeneous multi-agent systems including UGV and UAV”(ID: entropy-1890420). Those comments are all valuable and very helpful for revising and improving our paper, as well as the important guiding significance to our researches.Special thanks to you for your good comments. I have made modifications in Word version author-coverletter-201907171.Please see the Word.

Reviewer 3 Report

The presented manuscript deals with an interesting topic related to a Collaborative optimal formation control for heterogeneous multi-agent systems including UGVand UAV. The authors proposed a novel multiview representation formation control protocol, the distributed optimal formation control protocol, the cooperative formation control protocol.

This study certainly is interesting and contains a lot of new research and information based on conducted research.

Nevertheless, I have a few comments on the article:

 In the conclusions, you can add a few sentences that describe the results in more detail.

Did the data used in the research be complete and accurate?

In the simulation result section, the scheme of Matlab/Simulink should be described.

Are there limitations to the use of your method? If so, it would be worth mentioning in the conclusions?

Please also provide  some challenges and directions for future research. 

Author Response

Thank the reviewers’comments concerning our manuscript entitled “Collaborative optimal formation control for heterogeneous multi-agent systems including UGV and UAV”(ID: entropy-1890420). Those comments are all valuable and very helpful for revising and improving our paper, as well as the important guiding significance to our researches.Special thanks to you for your good comments. I have made modifications in Word version entropy-1890420-cover letter.Please see the Word.

Round 2

Reviewer 1 Report

The paper has been improved. However, there are still several issues. First, the simulation results are of toy size. Only three UAVs and three UGVs are considered. More complicated situations are needed. Otherwise, the theoretical results cannot be sufficiently verified. Secondly, the language needs to be improved. For example,' In an undirected graph, the number of vertices in which vertex  can exchange information is called degree , is the degree matrix of the graph, and is the sum of all elements in the row  of the adjacency matrix' The degree matrix is incorrectly defined. I recommend a professional editing service. Finally, the third author's name is displayed using a different font. This is very suspicious. Please clarify it. 

Author Response

Thank the reviewers’comments concerning our manuscript entitled“Collaborative optimal formation control for heterogeneous multi-agent systems including UGV and UAV”(ID: entropy-1890420). Those comments are all valuable and very helpful for revising and improving our paper, as well as the important guiding significance to our researches.Special thanks to you for your comments. We appreciate for reviewer, and hope that the correction will meet with approval.Once again, thank you very much for your comments .

Please give some advice on docx 'author-coverletter-22558643.V2'.

Reviewer 2 Report

My recommendation is to accept the article.

Author Response

Thank the reviewer for accepting the manuscript (ID:entropy-1890420).Thank the reviewers’comments concerning our manuscript entitled “Collaborative optimal formation control for heterogeneous multi-agent systems including UGV and UAV”(ID: entropy-1890420). Thank you very much for your work and your approval of the revised version of our paper.It is because of the comments of the reviewers that our paper has been greatly improved.On behalf of my co-authors, we would like to express our great appreciation to reviewers.

Round 3

Reviewer 1 Report

The paper can be accepted.